# Physiological, Agronomical, and Proteomic Studies Reveal Crucial Players in Rice Nitrogen Use Efficiency under Low Nitrogen Supply

**DOI:** 10.3390/ijms23126410

**Published:** 2022-06-08

**Authors:** Aadil Yousuf Tantray, Yehia Hazzazi, Altaf Ahmad

**Affiliations:** 1Department of Botany, Aligarh Muslim University, Aligarh 202002, Uttar Pradesh, India; a.y.tantray@gmail.com; 2School of Biological Sciences, University of Aberdeen, Aberdeen AB24 3UU, UK; y.hazzazi.19@abdn.ac.uk; 3Biology Department, Faculty of Sciences, Jazan University, Jazan 45142, Saudi Arabia

**Keywords:** leaf proteome, nitrogen metabolism, nitrogen use efficiency, photosynthesis, rice

## Abstract

Excessive use of nitrogenous fertilizers to enhance rice productivity has become a significant source of nitrogen (N) pollution and reduced sustainable agriculture. However, little information about the physiology of different growth stages, agronomic traits, and associated genetic bases of N use efficiency (NUE) are available at low-N supply. Two rice (*Oryza sativa* L.) cultivars were grown with optimum N (120 kg ha^−1^) and low N (60 kg ha^−1^) supply. Six growth stages were analyzed to measure the growth and physiological traits, as well as the differential proteomic profiles, of the rice cultivars. Cultivar Panvel outclassed Nagina 22 at low-N supply and exhibited improved growth and physiology at most of the growth stages and agronomic efficiency due to higher N uptake and utilization at low-N supply. On average, photosynthetic rate, chlorophyll content, plant biomass, leaf N content, and grain yield were decreased in cultivar Nagina 22 than Panvel was 8%, 11%, 21%, 19%, and 22%, respectively, under low-N supply. Furthermore, proteome analyses revealed that many proteins were upregulated and downregulated at the different growth stages under low-N supply. These proteins are associated with N and carbon metabolism and other physiological processes. This supports the genotypic differences in photosynthesis, N assimilation, energy stabilization, and rice-protein yield. Our study suggests that enhancing NUE at low-N supply demands distinct modifications in N metabolism and physiological assimilation. The NUE may be regulated by key identified differentially expressed proteins. These proteins might be the targets for improving crop NUE at low-N supply.

## 1. Introduction

Nitrogen (N) is an essential component of nucleic acids, amino acids and proteins, chlorophyll, ATP, some metabolites, and specific hormones in plants for growth, development, and production [1,2,3]. Rice is a major cereal crop. Its demand is rising as the global population grows [4]. Currently, 3.5 billion people consume 748 million tonnes of rice, accounting for 15% of global applied N [5]. Assuming a world population of 10 billion by 2050 [6], the N demand for rice production will rise by 44%. Since crop plants only utilize 30–40% of applied nitrogen, the remaining 60–70% of unused N in agriculture is causing severe environmental and health problems [7]. The unused soil N volatilizes into the air as pollutants such as ammonia and nitrogen oxides, harming ecosystems through ozone depletion, eutrophication, and soil acidification; evidence of these harmful processes can be found in dead zones in the Gulf of Mexico [8,9]. The unused soil N also impacts the economy [10]. To reduce resource waste and pollution and achieve sustainable development, optimizing N use is critical. This can be achieved by selecting or breeding high NUE cultivars for large-scale cropping systems [10,11,12].

Crop NUE is often defined as seed yield per unit of N [13]. Many factors contribute to effective N use for seed development, including soil uptake, root-to-shoot N translocation, assimilation in source organs, and subsequent allocation of primarily amino acids to seed sinks [13,14,15]. Plants store N as nitrate, amino acids, or proteins during vegetative growth in above-ground tissues [16,17]. The stored N is remobilized for growth and storage product accumulation in seed sinks [18,19]. The stress-induced breakdown of functional proteins can also produce remobilized N, especially Rubisco or organelles [20,21,22]. This depends on crop species, genotypes, soil, and environmental conditions [11,23,24]. Both N uptake and utilization contribute to NUE and rice cultivars from different genetic backgrounds differing in N source-sink ratio, and this process has an effect on physiological efficiency [25].

In plants, photosynthesis is the crucial physiological process for growth and development [26], and its efficiency is influenced by N application [27]. One part of the absorbed N forms a Rubisco protein, and the second part is used in other photosynthetic components [28]. To overcome the dysfunction of photosynthetic components, remobilization of leaf N content (LNC) plays a vital role [29]. Photosystem II (PSII) is the main component of this process, which regulates electron transport flow and, thus, helps generate assimilatory powers in the form of ATP and NADPH [30]. Nitrogen plays a significant role in regulating photochemical quantum yield and quenching efficiency of PSII, and its limitation causes photoinhibition in rice plants [31,32,33,34]. In addition, the Rubisco protein, gas exchange, and chlorophyll fluorescence reveal a significant correlation among traditional rice varieties [34,35]. Furthermore, the function of these physiological traits directly impacts plant growth and crop productivity [36].

In rice and other cereal crops, N is assimilated in roots into amino acids transported to the shoot through the xylem, or nitrate may be assimilated in leaves while assigned through the transpiration stream [37]. During early growth stages, most of the assimilated N may be used for leaf metabolism or storage [38,39], while at higher growth stages, N mainly functions for the tillering and synthesizing of amino acids for grain development [36,40]. The primary sources for N re-distribution to grains are leaves, which constitute ~20% of the total N content [36,41,42]. Among different developmental growth stages, the flag leaf has been identified as that which produces the most carbohydrates in plants [12] and as the most photosynthetically active compared with other leaves [43]. Therefore, carbon and N assimilation and their remobilization are crucial for NUE and grain production in plants. Furthermore, NUE variation has been conferred by genetic variants in rice [44,45], and the genetic basis for NUE remains unexplored [46].

Comparative analysis of the genetic basis of NUE in a few crop species has been mainly carried out under controlled environmental conditions in greenhouse or hydroponics system, and it has been primarily focused on a shoot or root function at the seedling or vegetative growth stages [47,48,49,50,51,52]. However, plant growth and production under controlled conditions vary significantly from the field conditions [53,54]. Moreover, plants focus on nutrient acquisition in vegetative tissues during the early growth stages. In contrast, the resources are re-allocated to the reproduction phase and grain production during later growth stages by different molecular and physiological processes [39,40,55]. The leaves of the vegetative and reproductive growth stages are the primary source of C and N supply for physiological efficiency and grain development; we hypothesized that multifaceted tunings, especially in C assimilation and N metabolism and distribution, would require enhancements in NUE and grain production in plants. It was further predicted that rice cultivars possess physiological and agronomic efficiency differences due to existing genetic variation for NUE. Two contrasting rice cultivars with different growth and physiological efficiency were selected and grown in field conditions under optimum and low-N supply conditions, and these were investigated during different growth stages. We explored physiological, developmental, and proteome responses of the two rice cultivars and identified crucial molecular mechanisms in specific growth stages prompting differences in NUE and crop yield under low-N supply.

## 2. Results

### 2.1. Selection of Two Contrasting Rice Cultivars

At the fifth tiller stage, physiological, growth, and biochemical differences were measured in ten rice cultivars under low N, P, and S (NPS) conditions. All the traits measured except transpiration rate (*p* = 0.174) and momentary fluorescence (*p* = 0.197) have shown significant differences between the rice cultivars (Appendix A). Gas-exchange parameters like net photosynthetic rate (P_n_), stomatal conductance (g_s_), intercellular CO_2_ concentration (C_i_), and chlorophyll content measured by SPAD (soil and plant analyzer development) revealed significant variation between the rice cultivars (Appendix A). Also, a significant variation between the rice cultivars was observed in chlorophyll fluorescence parameters (Appendix A). In addition, plant growth parameters like shoot and root length, leaf area, and plant biomass depicted great genetic variation in the cultivars for low NPS supply (Appendix A). Similar observations were given by Rubisco activity and soluble protein content (Appendix A). There was a significant variation in nutrient uptake efficiency between the rice cultivars, depicted by leaf N, P, and S contents (Appendix A). Principle component analysis simplifies the data sets by correlating the responses and revealing Panvel as high-efficient and Nagina 22 as low-efficient cultivars among the ten rice cultivars (Figure 1). The growth and gas-exchange parameters have shown a strong correlation with nutrient uptake (dominated in group I and IV), while non-photochemical quenching (NPQ) (in group II) have shown a weak correlation with photochemical quenching (qP) (in group III). The two contrasting rice cultivars (Panvel and Nagina 22) were selected for further stage-specific experiments based on the above analysis.

### 2.2. Analysis of Variance of the Stage-Specific Experiment at Low-N and Optimum-N

Analysis of variance (ANOVA) of gas-exchange, chlorophyll fluorescence, biochemical, and plant growth traits was held with a 3-way ANOVA model, using cultivars (C) and treatment (T) as fixed factors and growth stage (GS) as a random factor (Appendix A). All parameters measured in the experiment were significant, but traits such as F_v_/F_m_, NPQ, shoot length, plant biomass, Rubisco enzyme activity, and LNC exhibited a strong significant difference between treatments and cultivars. Interactions between factors C × T and T × GS were significant in most of the traits, whereas C × T × GS interaction was significant only in a few traits such as P_n_, F_v_/F_m_, ΦPSII, ETR, NPQ, shoot length, plant biomass, Rubisco enzyme activity, and LNC. Agronomic and NUE parameters were statistically analyzed with a 2-way ANOVA model that treated cultivars (C) and treatments (T) as factors (Appendix A). All parameters were significant at C and T factors; besides, traits like total tillers, grain yield, NUpE, NUtE, and NUE have shown strong significant variation. A significant C × T interaction was observed in traits except for the 1000 grains weight parameter.

### 2.3. Panvel and Nagina 22 Show Physiological Differences at Low N and Optimum N Supply

Gas-exchange traits (P*_n_* and g*_s_*) and chlorophyll content (SPAD) were measured at six growth stages in cultivars Panvel (high-efficient) and Nagina 22 (low-efficient) at 100% and 50% recommended dose of N supply (Figure 2). The P*_n_* was significantly (*p* = 0.001) decreased at panicle and milk stages by 16.68% and 19.59%, respectively, in Panvel and at 6th tiller, flag leaf, booting, panicle, and milk stages by 21.89%, 21.10%, 20.93%, 23.45% and 25.29%, respectively, in Nagina 22 under N-50% rather than N-100% treatment (Figure 2A). The g*_s_* decreased significantly (*p* = 0.003) by 10.03% and 13.64% at panicle and milk stages, respectively, in Panvel and by 12.40%, 11.98%, 16.36%, and 18.49% at flag leaf, booting, panicle, and milk stages, respectively, in Nagina 22 plants grown under N-50% treatment rather than the N-100%-treated plants (Figure 2B). The chlorophyll content was increased from the 3rd tiller to panicle under both the treatments in both cultivars (Figure 2C). Booting, panicle, and milk stages have shown a significant (*p* = 0.01) decrease of 11.54%, 18.20%, and 21.26%, respectively, in the chlorophyll content in Nagina 22 under N-50% treatment. In contrast, in Panvel, it decreased significantly by 9.39% and 10.98% at panicle and milk stages, respectively, under N-50% conditions as opposed to N-100%-treated plants.

Chlorophyll fluorescence efficiency (F_v_/F_m_, ΦPSII, and ETR) was decreased in rice plants cultivated under N-50% treatment rather than in the plants grown under N-100% treatment (Table 1). The F_v_/F_m_ decreased significantly (*p* = 0.001) by 11.5–12.8% between booting to milk stages in Panvel, but in Nagina 22, it decreased by 7.5–18.4% at all growth stages under N-50% rather than the N-100% treatment. The ΦPSII decreased significantly (*p* = 0.008) in both cultivars under N-50% treatment rather than N-100% treatment; the ΦPSII decreased by nearly 10% from flag leaf to milk stages and by 11.8–20.6% from 6^th^ tiller to milk stages in Panvel and Nagina 22, respectively. Also, the ETR decreased according to the ΦPSII in plants grown under N-50% conditions rather than N-100% treatment in both rice cultivars. The rice leaves*’* quenching (qP and NPQ) efficiency varied between N applications (Table 1). Under N-50% rather than N-100% treatment, the qP decreased by 8–13.6% between flag leaf to milk stages in Panvel, and by more than 10–25% at all growth stages except 3^rd^ tiller in Nagina 22. In turn, the NPQ increased by 20% and 23.2% at panicle and milk stages, respectively, in Panvel, and by 24–40% at all growth stages in Nagina 22 under N-50% conditions compared to the N-100%-treated plants of the respective cultivars.

### 2.4. Plant Growth and LNC Vary between Rice Cultivars at a Low-N Supply

Plant height, leaf area, plant biomass, and LNC were measured in Panvel and Nagina 22 cultivars under 50% and 100% N supply (Figure 3). Between cultivars, the plant height of Panvel was more than the Nagina 22 at both N-50% and N-100% treatments. However, between treatments, the plant height decreased significantly (*p* = 0.001) by ~23% between flag leaf to milk stages under N-50% compared to the results observed in the plants grown under N-100% in Nagina 22 (Figure 3A). In cultivar Panvel, the plant height decreased significantly by 11% at panicle and milk stages under N-50% treatment as opposed to the results observed in N-100%-treated plants. The leaf area increased during vegetative growth, and significant differences (*p* = 0.007) were observed under N-50% conditions, where it by decreased ~13% at flag leaf to milk stages in Nagina 22 and by 9% at panicle and milk stages in Panvel compared to the results observed in the plants grown under N-100% treatment (Figure 3B). The plant biomass significantly (*p* = 0.005) decreased under N-50% treatment by ~23% and ~12% from flag leaf to milk stages in Nagina 22 and Panvel, respectively, compared to results observed in the N-100%-treated plants (Figure 3C). Similarly, the LNC significantly (*p* = 0.001) decreased by 10–13% in Panvel and by 11–32% in Nagina 22 at different growth stages under N-50% treatment compared to the the plants grown under N-100% treatment (Figure 3D).

### 2.5. Rubisco Activity and Soluble Protein Differ between Panvel and Nagina 22

Rubisco activity and soluble protein content in the leaves of rice decreased under low N application compared to the optimum N application at six growth stages in both cultivars Panvel and Nagina 22 (Table 2). Rubisco enzyme activity (*p* = 0.001) decreased by ~12% and ~20% from flag leaf to milk stages under N-50% treatment in Panvel and Nagina 22, respectively, compared to the plants grown under N-100% treatment. The same was reflected in Rubisco protein content (*p* = 0.009), which decreased from flag leaf to milk stages by ~10% in Panvel and by ~15% in Nagina 22 under N-50% compared to N-100%-treated plants. Like viz, the soluble protein content (*p* = 0.004) of the leaves was ~11% (in Panvel) and ~21% (in Nagina 22), decreasing at higher growth stages in N-50% treated plants compared to N-100%-supplied plants.

### 2.6. Agronomic Parameters and N Use Efficiency Are Enhanced in Panvel

The agronomy efficiency of plants decreased in both rice cultivars, Panvel and Nagina 22, under low-N supply than in the plants grown under optimum-N supply (Figure 4). The total tiller number per hill (*p* = 0.001) decreased by 18.85% in Panvel and by 31.52% in Nagina 22 under low N compared to optimum-N-treated plants (Figure 4A). Panicle production (*p* = 0.000) decreased in both the cultivars under low-N supply; panicle number per hill was 14.02% and 26.23% lesser, respectively, and panicles per meter square were 14.02% and 26.28% lesser in Panvel and Nagina 22, respectively, than in the plants treated with optimum N (Figure 4B,C). At low-N supply, panicle length decreased by 8.04% and 18.52% in Panvel and Nagina 22, respectively (Figure 4D). Low N application decreased grain set in panicles; the effect was more severe in Nagina 22 (33.33%) than in Panvel (15.48%) rice cultivar (Figure 4E). The grains (*p* = 0.001) in rice cultivated under low N treatment were 10% (Panvel) and 12.70% (Nangina 22) lighter in weight than the grains under optimum N treatment (Figure 4F). Finally, grain yield (*p* = 0.000) was measured and showed a severe decrease in cultivar Nagina 22 (57.28%) and in Panvel (35.04%) at low-N supply when compared to plants grown under optimum N supply (Figure 4G).

Nitrogen uptake efficiency (NUpE), N utilization efficiency (NUtE), and N use efficiency (NUE) were calculated after harvesting the crop and were improved in low N treated plants as compared to optimum N-treated plants in both rice cultivars. The NUpE increased by 13.82% more in Panvel than in Nagina 22 under low-N supply (Figure 4J), whereas the NUtE increased by 8.7% more in Panvel than in Nagina 22 (Figure 4K). The overall NUE improved in Panvel by 13.22% more than in Nagina 22 under low N-supplied plants; with regard to NUE, both cultivars grown under low-N conditions exhibited significant differences in comparison to the optimum N-supplied plants (Figure 4L).

### 2.7. Differentially Expressed Proteins in Panvel and Nagina 22 Rice Cultivars

Leaf proteomes of the six growth stages of Panvel and Nagina 22 were evaluated with two-dimensional gel electrophoresis (2-DE) at 50% and 100% N supplies (Appendix A). After staining the gels, more than 550 reproducible spots were detected in the 2-DE maps. Although several proteins showed differential expression, only 65 exhibited a two-fold change in their abundance during the experiment’s six growth stages (Figure 5A). Among these proteins, 49 (75%) were downregulated while 16 (25%) were upregulated at low-N supply. Several differentially expressed proteins (DEP) varied between the rice cultivars at different growth stages (Figure 5B). The number of downregulated proteins was higher in Nagina 22 at early growth stages such as 3rd tiller (15), flag leaf (13), and booting (17) stages. The highest number of commonly downregulated proteins in both cultivars was observed at the milk stage (44). A higher number of upregulated proteins was observed in Panvel at higher growth stages, such as flag leaf (5), panicle (5), and milk (6) stages. The number of commonly upregulated proteins in both cultivars was highest (12) at the 3rd tiller and booting stages.

### 2.8. Spatial Distribution and Cellular and Molecular Functions of DEPs

Differentially expressed proteins were localized in different cellular components (Figure 6A). Among the cellular components, higher DEPs were contributed by chloroplast (33%), followed by cytosol (21%) and mitochondrion (18%), whereas the lowest DEPs were contributed by endoplasmic reticulum (3%). Based on the biological function (Figure 6B), most of the proteins were involved in electron transport (11.48%), followed by oxidative stress (8.20%), protein folding/maturation (8.20%), Calvin cycle (6.56%) and nitrogen assimilation (6.56%). In contrast, few proteins were involved in carbon utilization (4.92%), transcription (3.28%), and ATP synthesis (1.64%). Based on the molecular function (Figure 6C), many proteins were involved in an ATP binding activity (13.73%), followed by transferase activity (7.84%), and metal-ion binding (7.84%), and chlorophyll-binding (5.88%). In contrast, some proteins were involved in activities like chaperone (3.92%), RNA binding (3.92%), RuBisCO activator (1.96%), and others. The detailed description of DEPs and their theoretical and experimental molecular weight and isoelectric point, chromosome number, mode of regulation, function, and expression level at different growth stages are given in Figure 6D, Appendix A.

### 2.9. Proteins Involved in Photosynthesis and Carbon Assimilation Are Downregulated in Nagina 22

Proteins differentially expressed in Panvel and Nagina 22 at different growth stages under low-N supply and associated with C and N metabolism and other processes were analyzed (Figure 6D, Appendix A). In the photosynthesis process, DEPs involved in PSII function such as the photosystem II reaction center PSB28 protein, theoxygen-evolving enhancer protein 3, and the chlorophyll a-b binding protein were significantly downregulated in Nagina 22. In addition, a protein function for electron transfer between PSII and PSI, cytochrome b6-f complex iron-sulfur protein, was downregulated in Nagina 22. Further, proteins associated with PSI such as photosystem I reaction center subunit VI and photosystem I P700 chlorophyll a apoprotein A1 were also significantly downregulated in Nagina 22 (Figure 6D). Other proteins related to the light-reaction of photosynthesis, such as plastocyanin, ferredoxin-1, and ferredoxin-NADP reductase leaf isozyme 1, were highly downregulated in Nagina 22. Even chlorophyll synthase was significantly more reduced in Nagina 22 than in Panvel. Energy and electrons are produced in light reactions for photosynthetic C assimilation in the Calvin cycle with the help of the Rubisco enzyme. Proteins of the Rubisco large chain and small subunit A, together with Rubisco activase, were significantly more reduced in Nagina 22 than in Panvel (Figure 6D), revealing that Panvel channels more N into its photosynthetic apparatus for improved C fixation. In addition, proteins involved in C metabolism such as fructose-1,6-bisphosphate, phosphoribulokinase, and pyruvate dehydrogenase E1 component subunit alpha-1 were downregulated in Nagina 22. However, enzymes like malate dehydrogenase and phosphoenolpyruvate carboxylase were reduced, and carbonic anhydrase and enolase were elevated in Pavel (Figure 6D), pointing to more energy stabilization in Panel compared with Nagina 22.

### 2.10. Variation in Differential Expression of Protein Turnover, N Metabolism, and Stress Related Proteins

Proteins involved in amino acid synthesis apparatus and metabolisms such as the 40S ribosomal protein S21 and the 60S ribosomal protein L18, together with aspartate aminotransferase and methionine aminopeptidase, were more reduced in Nagina 22 than in Panvel (Figure 6D, Appendix A). In addition, proteins involved in translation and post-translational modification, such as elongation factor Tu and 60kDa chaperonin alpha subunit, were downregulated in Nagina 22. In contrast, protein disulfide isomerase-like 1–1 and chaperonin were upregulated in Panvel, probably for protein stabilization.

Nitrogen metabolism was prominently more reduced in Nagina 22 than in Panvel due to the downregulation of associated proteins. Proteins such as the nitrogen regulatory protein *P*-II homolog, nitrate reductase [NADH] 1, and nitrogen fixation protein were significantly reduced in Nagina 22 (Figure 6D, Appendix A). Other proteins involved in the associated biological process, such as amine oxidase and ATP synthase, were upregulated in Panvel. In addition, proteins involved in the transportation of ions such as H(+)-exporting diphosphatase and potassium channel AKT2 were significantly elevated in Panvel.

Commonly oxidative stresses are generated at a low nutrient supply, which is overcome by the anti-oxidative defense in plants. Proteins such as glutathione reductase, glyoxalase, and peroxidase were significantly upregulated in Panvel compared with Nagina 22 (Figure 6D, Appendix A). In addition, other proteins involved in stress management, such as abscisic stress-repining protein 5, were upregulated in Nagina 22, and RUBP-domain-containing protein 6 was upregulated in Panvel. Other exciting proteins were reported for the first time; protein chloroplast stress enhancing tolerance and clp amino terminal domain containing protein were significantly reduced in Nagina 22 compared to Panvel (Figure 6D, Appendix A).

### 2.11. Validation of Gene Expression of Some Proteins by Quantitative Real Time-PCR

Ten individual genes were selected to validate gene expression levels by quantitative Real Time-PCR (qTR-PCR) with gene-specific primers and Actin 1 as a reference gene. The relative expression of the selected genes was evaluated in independent replicates at the six specific growth stages in both Panvel and Nagina 22 cultivars. The expression level of the selected genes was higher than their protein level (Figure 7). Among these genes, Os011g0639900 (carbonic anhydrase) was upregulated in both rice cultivars, but the fold-expression was eight times more in Panvel than in Nagina 22. The other nine genes were downregulated. RCA, CHLG, and Os11g07916 (nitrogen fixation protein) are more severely affected in the Nagina 22 than in the Panvel rice cultivar, but NIA1, PSB28, and GST are more affected in the Panvel than in the Nagina 22 cultivar (Figure 7A,B). The GLB, PSAH, and RCABP89 have displayed almost the same level of down regulated expression in both rice cultivars.

## 3. Discussion

Nitrogen is essential for plant growth and development [56]. Physiological differences were observed between the rice cultivars under low nutrient supply (Appendix A). Based on the physiology and growth differences between the rice cultivars, a pair of rice cultivars, Panvel (high-efficient) and Nagina 22 (low-efficient), were selected for long-term stage-specific experiments (Figure 1). Phenotypic differences were observed at different growth stages between the rice cultivars Panvel and Nagina 22 under low-N supply (Figure 2 and Figure 3; Table 1 and Table 2). Nagina 22 plants showed a significantly decreased photosynthetic rate, stomatal conductance, and relative chlorophyll content at higher growth stages (Figure 2), which may be due to the decreased fluorescence efficiency of PSII with limiting N availability for regulation of photosystems and chlorophyll molecules. The severe impact of low-N supply was observed on F_v_/F_m_, ΦPSII, and ETR in Nagina 22 than in Panvel plants, and quenching efficiency (NPQ) suggested that Nagina 22 dissipated more light photons into heat energy due to higher values of NPQ, notably at reproductive growth stages (Table 1). However, a decrease in photosynthesis was reported at a single growth stage under low-N supply in earlier studies [57,58,59]. We confirmed it at multiple vegetative and reproductive growth stages in rice plants. Proper N-fertilization cannot alone improve the photosynthesis rate [60]. However, the maintenance of N-dependent photosynthetic components is necessary during the growth of rice plants [61]. Fluorescence measures the efficiency of photosystems non-invasively in plants [62]. The malfunction of PSII due to low-N supply was studied, where increased NPQ explained the dissipation of exciting energy as heat [63] and slow rate of the PSII antennae recovery from quenched to unquenched state [64]. Furthermore, a significant decrease in ETR was observed in Nagina 22 as compared to the Panvel, maybe due to low proton motive force during the photochemical reaction [65] and, thus, transiently hindering the photosynthesis in Nagina 22, particularly at reproductive growth stages [66].

The plants depicted a decrease in photosynthesis and fluorescence; this decrease affected other aspects of growth, such as plant height, leaf area, and plant biomass, which decreased more in Nagina 22 than in Panvel under low-N supply (Figure 3A–C). The variation in growth parameters at different growth stages was very similar to the leaf N status of the plants (Figure 3D), which suggests lower N regulation in Nagina 22 plants than in Panvel ones. However, some studies suggested that an increase in photosynthesis parameters at a specific growth stage is maybe less relevant for biomass and yield improvements in some crops [67,68], but in rice, enhancing photosynthesis in a single leaf would increase yields [58]. The photosynthesis and plant-growth traits positively correlated with the leaf N status at multiple growth stages in the present study. In addition, Murata [69] has also found a strong positive correlation between leaf photosynthesis and crop-growth rates in many crops. Further, the increase of LNC per unit increases the rate of photosynthesis and biomass production in rice [58,70], which may be caused by greater N allocation to Rubisco [58,71]. Rubisco functions as the primary CO_2_ fixation enzyme, and the properties and amount of this enzyme significantly affect the photosynthetic rate [58]. Our study also showed that the enzyme activity and amount of Rubisco were decreased under low-N supply (Table 2) and strongly correlated the photosynthesis and growth assays of rice plants. Along with the Rubisco activity, the soluble protein content of the leaf was also severely decreased in Nagina 22 compared to Panvel under low-N supply (Table 2).

Improvement in the yield of cereal crops depending on a wide scale of N fertilizer inputs has been observed globally. Agronomic traits like grains per unit area, the proportion of grains fill, and grain weight determined the cereal yield in crops [58]. In our study, the grain yield per unit land area was severely declined in Nagina 22 as compared to Panvel under low-N supply (Figure 4G), which was due to the low production of tillers per hill, panicles per unit land area, and grains per panicle (Figure 4A–E) in Nagina 22. In rice, the number of tillers is determined by inputs of N fertilization [72], and productive panicles and grain number have proven to be a limiting factor in rice yield [73,74], both of which are affected by N supply. The low impact in grain yield for Panvel at low-N supply suggests that this cultivar has more efficiency in N-uptake and utilization for grain growth than Nagina 22. In fact, the leaf N levels in Panvel were increased by up to ~20% at higher growth stages when compared with the Nagina 22 at low-N supply (Figure 3D). The overall result was an enhanced NUE in Panvel (Figure 4L) due to increased NUpE and NUtE for low-N supply than Nagina 22 (Figure 4J,K). These outcomes suggest that the influence of NUpE and NUtE on NUE strongly differs dependent on the amount of N fertilization and rice cultivar [25]. While other studies in cereals have also demonstrated that the NUpE influences NUE at low-N supply [75,76], and NUtE was more closely associated with genetic variation in NUE under different N regimes rather than NUpE [58,77,78]. Indeed, in the current study, growth and yield enhancements in Panvel versus Nagina 22 were detected under the low-N supply (Figure 3 and Figure 4), and NUE was more improved due to NUpE and NUtE in Panvel than in Nagina 22 (Figure 4J–L). Together, this underpins that the high-efficient Panvel and low-efficient Nagina 22 represent exceptional candidates for studying genetic differences concerning rice yield and NUE.

Higher N uptake and utilization in Panvel than in Nagina 22 plants under low-N supply resulted in enhanced N allocation to leaves (Figure 3D and Figure 4J,K), as supported by higher protein levels in leaves at different growth stages (Table 2). In addition, the improved leaf N status in Panvel positively affects photosynthesis (Figure 2A–C), fluorescence (Table 1), growth assays by C fixation, and assimilation of specific-growth stages of rice plants (Figure 3A–C). Our results align with earlier studies on Arabidopsis, rice, and wheat, studies showing significant N allocation to leaves promotes plant growth and C fixation in leaves [36,56,58]. These studies further suggest that improved N partitioning in leaves affects the photosynthetic light and dark reactions, which were armored by improved chlorophyll contents, electron transport rate, enzyme activity, and Rubisco amount. In fact, our proteome analyses depicted a downregulation of perceptive proteins associated with the light reaction of photosynthesis and Rubisco regulation in leaves of specific-growth stages of Nagina 22 compared with Panvel under low-N supply (Figure 6). Some of those proteins function for photosynthesis and plant growth, including chlorophyll a-b binding protein, cytochrome b6/f complex iron-sulfur subunit, FNRs, PSII oxygen-evolving enhancer protein, PSI and PSII subunits, and Rubisco large and small chains have earlier been reported [36,79,80,81,82], and supports our premise that the identified proteins also play a pivotal role in NUE under low-N supply. In addition to these proteins, chlorophyll synthase involved in chlorophyll biosynthesis and Rubisco activase regulating Rubisco enzyme during C fixation in dark reaction were more downregulated in Nagina 22 than in Panvel leaves under low-N supply [28]. The overall downregulation of these proteins was detected in higher growth stages (Figure 5B and Figure 6D), which may cause a prominent effect on photosynthesis and growth in the reproductive phase of Nagina 22 plants under low-N supply (Figure 2 and Figure 3). Schematic representation of some important DEPs shows proteins’ coordinative functions in light and dark reactions of photosynthesis and other biological processes for N-efficient model plants (Figure 8).

Further, our data suggest that improvements in rice NUE under low-N supply require regulation of proteins involved in C metabolism, energy stabilization, and source-to-sink transport as supported by the upregulation of carbonic anhydrase and isocitrate dehydrogenase in Panvel and the downregulation of amino methyltransferase, ATP-citrate synthase, and fructose-1,6-bisphosphate in Nagina 22 (Figure 6D). Our findings are supported by preceding studies that have considered these proteins as low-N responsive proteins in rice [83,84], maize [85], tea [86], and wheat [36]. Differential expression of these proteins at both vegetative and reproductive growth stages (Figure 5B and Figure 6D; Appendix A) suggests that modifying such genes may successfully improve NUE and seed yield under low-N supply [36,84]. Moreover, H(+)-exporting diphosphatase and potassium channel AKT2 were detected as downregulated in Nagina 22 and as upregulated in Panvel (Figure 5B and Figure 6D; Appendix A), functioning for source-to-sink transportation and would be interesting for improving NUE, which remains to be determined in the future.

The involvement of N metabolism for NUE began with the uptake of ammonium and nitrate from the soil, and then either the reduction of the N to amino acids in the root before being transported to the shoot or the allocation of nitrate to leaves where it is integrated into GS/GOGAT assimilatory pathway [37,87]. The critical enzyme of nitrate metabolism is nitrate reductase, and nitrogen regulatory protein P-II interacts with glutamate kinase to regulate amino acid biosynthesis [88], which were upregulated in Panvel and downregulated in Nagina 22 (Figure 6D). In addition, an ammonium assimilation key enzyme, glutamate synthase involved in glutamate biosynthesis, and nitrogen fixation protein were downregulated in Nagina 22 compared to Panvel (Figure 6D; Appendix A). The involvement of these proteins in developing tillers [89] suggests good candidate genes for improving NUE and crop yield in rice plants. In fact, the overexpression of GS/GOGAT genes has shown to be a great success in improving NUE or seed yield under a range of N supply [90,91,92]. Indeed, our proteomic study also displayed the differential expression of proteins in Panvel rather than in Nagina 22, and this result is associated with transcription, translation, and protein folding molecular processes such as ribosomal proteins, elongation factor, RNA-binding proteins, and chaperonins (Figure 6D; Appendix A). These proteins play an important role in amino acids and protein biosynthesis, as well as in protein functioning at plants’ vegetative and reproductive growth phases [93,94,95]. Furthermore, our study detected the differential expression of two novel proteins at most of the growth stages under low-N supply, (i) the downregulation of CHLOROPLAST ENHANCING STRESS TOLERANCE in Nagina 22 that functions for chloroplast organization [96], and (ii) the upregulation of RUBP domain-containing protein in Panvel having an unknown function (Figure 6D; Appendix A). Besides photosynthesis and protein turnover, plants can produce reactive oxygen species (ROS) under N deficiency [97], affecting their growth and development [98]. To maintain an oxidative defense system, our study reveals significantly higher upregulation of glutathione reductase, peroxidase, and superoxide dismutase in Panvel than in Nagina 22 (Figure 6D), suggesting that Panvel has better adaptation for scavenging reactive species under low-N supply [85,99,100].

## 4. Materials and Methods

### 4.1. Plant Materials and Growth Conditions

Ten commonly cultivated rice cultivars (MTU-1010, Rasi, Nagina 22, BPT-5204, Pusa-44, Nidhi, Panvel, CR Dhan-310, CR Dhan-311, and Taipe 309), having diverse physiological and agronomic potentials, were procured from the Division of Agronomy, Indian Agricultural Research Institute, New Delhi, India. The ten rice cultivars were screened for low nutrient supply, and the growth and physiology of the plants were measured in field conditions. A pair of rice cultivars (Panvel and Nagina 22) having contrasted physiological and growth efficiencies (Appendix A) were selected for a stage-specific experiment under low and optimum N supplies in the field conditions. On the one hand, Nagina 22 is a heat and drought resistant rice cultivar with short, bold, and white grains; belongs to aus subspecies; and originated from India (https://rice-genome-hub.southgreen.fr/, accessed on 30 June 2022). On the other hand, the cultivar Panvel is resistant to blast and salinity; has short, bold, and high grain yield; belongs to an indica subspecies; and originated from India (https://dbskkv.org/Research/Varieties.html, accessed on 30 June 2017).

To screen the rice cultivars and determine the stage-specific genetic and other physiological and agronomic differences, the rice cultivars were grown for two seasons from July–October (2017—screening; 2018—stage-specific experiments) in experimental fields of Agricultural Sciences, Aligarh Muslim University, UP, India. Aligarh (UP) is in a temperate climate zone, and the average temperature ranges from 24–34 °C (2017) and 25–35 °C (2018), average humidity was 51% (2017) and 53% (2018), and the average precipitation/rainfall was 112 mm (2017) and 116 mm (2018) during the crop seasons. The field experiments were carried out with a split-plot design with three replicates as subplots (6 m^2^). The fields were flood irrigated, and the water level was maintained more than 5 cm above the ground surface. The transplanted rice culture method was adopted. The seeds were sterilized before germination with 0.1% mercuric chloride for three min and then washed with distilled water five times. Sterilized seeds were sowed in nursery beds, and 25-day-old plantlets were transplanted in the subplots with a spacing of 10 cm × 25 cm between hills, and three plants per hill were planted. There were 40 hills m^−2^, and the plant density was ~120 plants m^−2^.

The agricultural field soil contained 1.02 g N kg^−1^ soil, 28.22 mg P kg^−1^ soil, and 14.31 mg S kg^−1^ soil. In the screening experiment, plants were fertilized with 50% of basal recommended doses (RD) of N, P, and S in the form of urea (46.4% N), diammonium phosphate (46% P), and potassium sulphate (18% S). For the soil, the RD of N is 120 kg ha^−1^, P is 60 kg ha^−1^, and S is 45 kg ha^−1^ [34,101,102]. The stage-specific experiment was supplied with two treatment levels, low N (50% RDN) and optimum N (100% RDN), and the treatments were given in three split doses, 50% at basal (before transplantation), 25% at maximum tiller stage, and 25% at the panicle stage (flowering). The P and S nutrients were supplied with 100% RD before transplantation in the field. In both experiments, other essential nutrients were provided before transplantation, such as potassium (potassium chloride) 40 kg ha^−1^ and zinc (zinc oxide) 25 kg ha^−1^.

In the screening experiment, plants of the fifth tiller growth stage were taken for physiology, growth, and biochemical analyses to select the contrasting rice cultivars. Six growth stages in the stage-specific experiment—3rd tiller, 6th tiller, flag leaf, booting, panicle, and milk stages—were taken for physiology, growth, and proteome analyses. At the harvesting time, some yield parameters were also measured in this experiment.

### 4.2. Analysis of Photosynthesis and Chlorophyll Fluorescence of PSII

Leaf gas-exchange parameters such as net photosynthetic rate (Pn) and stomatal conductance (gs) were measured in the morning (9:00–11:00 am) by an Infra-Red Gas Analyzer (CID-340, Photosynthesis system, Bio-Science, Camas, WA, USA). The average light intensity was ~800 μmol quanta m^−2^ s^−1^ during analyses. Leaf greenness or relative chlorophyll content was measured using a SPAD chlorophyll meter (SPAD 502 DL PLUS, Spectrum Technologies, Kakamigahara, Japan). Chlorophyll fluorescence of PSII of rice leaves was measured by a pulse amplitude modulation (Mini-PAM) chlorophyll fluorometer (Heinz Walz, Effeltrich, Germany). Before measurement, leaves were first dark-adapted for 30 min with leaf clips, and chlorophyll fluorescence was analyzed in saturation-pulse (SP) mode [34]. Dark-adapted leaves were induced with a transmutable light (<0.05 µmol m^−2^ s^−1^ for 1.8 µs) before SP to determine the minimum fluorescence yield (F_o_), and the maximum fluorescence yield (F_m_) was generated after saturating actinic light pulse (10,000 µmol m^−2^ s^−1^ for 0.6 µs). The fluorescence parameters were analyzed at both light- and dark-adopted states, using different calculations: (i) variable fluorescence (F_v_ = F_m_−F_o_), (ii) photochemical quantum efficiency of PS-II (F_v_/F_m_ = (F_m_−F_o_)/F_m_), (iii) non-photochemical quenching (NPQ = F_m_/F_m_’ − 1), and (ⅳ) electron transport rate (ETR = PPFD × 0.5 × abs I × ΦPSII) where PPFD = photosynthetic photon flux density, 0.5 = fraction of absorbed light allocated to PS-II, abs I = absorbed irradiance taken as 0.84 of the incident irradiance, and ΦPSII = effective photochemical quantum yield.

### 4.3. Estimation of Rubisco Activity and Soluble Protein Content of Leaf

Rubisco enzyme activity was determined by Usuda’s method [103], with a multi-mode microplate reader (Synergy H1, Biotek Instruments Inc., Pittsburgh, PA, USA) operating at 340 nm wavelength. Fresh 1 g leaf tissue was homogenized in a 0.25 M Tris-HCl extraction buffer (containing 0.0025 M EDTA, 0.05 M MgCl_2_, 2% DTT and pH 7.8). The extracts were subjected with 100 M Tris-HCl (pH 8.0) having reaction cocktail composition, 40 mM NaHCO_3_, 10 mM MgCl_2_, 5 mM DTT, 0.2 mM EDTA, 4 mM ATP, 0.2 mM NADH, 0.2 mM ribulose 1,5-bisphosphate, 1 U of 3-phosphoglycerate kinase, and 1 U of glyceraldehyde 3-phosphodehydrogenase, after centrifugation at 10,000× *g* at 4 °C for 10 min.

Soluble protein content was estimated by homogenized fresh-leaf material (0.5 g) in 0.1 M phosphate buffer (pH 6.8 at 4 °C) with the help of a pre-cooled mortar and pestle. The homogenate was transferred into 2.0 mL tubes and centrifuged at 5000× *g* at 4 °C for 10 min. The supernatant was transferred into a new 2.0 mL tube, and an equal volume of chilled 10% TCA was added for protein precipitation. The whole mixture was centrifuged at 3300× *g* at 4 °C for 10 min, the supernatant was discarded, and the remaining pellet was washed with cold acetone. The protein pellet was dissolved in 1.0 mL of 0.1 N NaOH in a 10 mL aliquot tube, and 5.0 mL of 1× Bradford’s reagent (Bio-Rad, Hercules, CA, USA) was added and vortexed. For optimal color development, the tubes were kept in the dark for 10 min, and the absorbance was measured at 595 nm by a microplate reader. The soluble protein content was estimated with the help of a standard curve using Bovine Albumin Serum (Sigma-Aldrich, St. Louis, MO, USA) as its standard and expressed in mg g^−1^ FW (fresh weight).

Rubisco protein was extracted using Carvalho’s method [104]. A 1 g fresh-leaf tissue was homogenized in a 0.2 M potassium phosphate extraction buffer (containing 50% (*w/w*) polyvinylpyrrolidone, 5 mM DTT, 5 mM EDTA, 0.2mM PMSF at pH 8.0). The supernatant was collected from homogenate after centrifugation at 27,000× *g* at 4 °C for 10 min, then desalted by Sephadex PD10 columns (Amersham Biosciences, Buckinghamshire, UK). The rubisco proteins were quantified with Bradford’s method.

### 4.4. Analysis of Growth and Yield Parameters

With randomly picked plants, plant-shoot length was measured by a meter scale (cm). Leaf area was measured by a standard method with a correction factor—(K): Leaf area (cm^2^) = K × leaf length (cm) × leaf breath (cm)—where K for rice leaves ranges from 0.67 to 0.80, but the value of 0.75 can be used for all growth stages except the seedling stage [105]. Plant biomass was measured after drying plants in an oven at 65 °C for 72 h, and the plant biomass was represented in g plant^−1^.

Crop yield parameters were measured in the stage-specific experiment on randomly picked plants at the time of harvesting when grains turned a golden color. The parameters measured are total tillers per hill, number of panicles per hill, panicles per meter square, panicle length (cm), and grains per panicle. The grains were dried in the open air for grain weight measurement at 14% grain moisture adjustment and represented in g 1000-grains^−1^. Grain yield per meter square was calculated and represented in kg m^−2^.

### 4.5. Analysis of Leaf NPS Contents and N Use Efficiency

Dried leaf samples were finely powdered, and a 5 mg leaf sample was used for estimation. The elemental analysis was performed by CHNS elemental analyzer (Vario EL III, Elementar Analyser Systeme, Germany) under the fundamental principle of the combustion process (furnace at 1000 °C); N is converted to N gas/oxides and S to S dioxide. The elements N and S were quantified with a thermal conductivity detector (set at 290 °C). For *p* estimation, the dried leaf samples were digested in the di-acid mixture (HNO_3_:HCl_4_ = 9:4 *v*/*v*). The total *p* content was determined in the digest spectrophotometrically at 625 nm, and the vanado-molybdate phosphoric acid yellow color method was used to make this determination [106]. The elemental content of N, *p*, and S of leaves were represented in mg g^−1^ DW (dry weight). NUpE, NUtE, and NUE were calculated according to Moll et al. [13]. NUpE was estimated as the ratio of total N content in the aboveground shoot/maturity time point to the total amount of N supplied. NUtE was estimated as seed yield relative to total N accretion in the aboveground shoot, and NUE was estimated as the ratio of seed yield to total N supply.

### 4.6. Proteome Analysis

Leaf samples of the specific growth stages were sampled in three biological replicates from the field and carried in liquid nitrogen to the laboratory. The leaf samples were lyophilized (Labcono, FreeZone 4.5 L, Kansas City, MO, USA) at −50 °C and vacuum pressure of 0.02 mBar for long-term storage. Fine powder of 0.5 g leaf material was prepared in liquid nitrogen, and total protein was extracted with the help of the ReadyPerpTM protein extraction kit (Bio-Rad, Hercules, CA, USA) following the manufacturer’s protocol. The protein pellet was harvested and solubilized in a cocktail buffer containing 7 M urea, 2 M thiourea, 50 mM DTT, and 4% CHAPS (Bio-Rad, Hercules, CA, USA, protein-grade), and protein concentration was estimated with Bradford’s reagent. For proteome profiling, two-dimensional electrophoresis (2-DE) was performed according to the method of O’Farrell [107]. An immobilized dry strip gel (11 cm, linear-gradient, pH 4–7) was rehydrated at 20 °C for 12 h in 180 μL of a sample containing 350 μg proteins [108]. The first dimension, i.e., isoelectric focusing, was accomplished by an isoelectric focusing (IEF) apparatus (PROTEAN^®^ i12™ IEF Cell, Singapore). Voltage programming of ProteinCell was set at 250 V for 30 min, 500 V for 30 min, 1000 V for 1 h, 2000 V for 1 h, followed by a linear increase of 6000 V to a total of 65.00 kVh. After completing the first dimension, the strips were subjected to reduction by an equilibration buffer with 130 mM DTT and then alkylation with 135 mM iodoacetamide. The second dimension, SDS-PAGE, was carried out for the separation of proteins with 12% SDS-PAGE in a vertical large format electrophoresis cell (PROTEAN^®^ II Xi Cell, Hercules, CA, USA) at a constant voltage of 150 V. Gels were stained overnight with Coomassie Brilliant Blue R-250 dye, and then the stain was removed with an acetic acid solution. Digital imaging of the gel was captured by a high-resolution scanner (HP Scanjet G3110), and image master PDQuest software (version 8.0, Bio-Rad, Hercules, CA, USA) was used for gel spot analysis. Optimized parameters were considered partial threshold 4, saliency 2.0, and minimum area 50. Quantification of spots was held based on their relative volume and quality, which was concluded by the ratio of the single spot volume to the whole comparative set of the spots. The spots in the analyzer manager with >2-fold change intensity in volume during the comparison or significant variation between the control and other treatments decided by the paired Student’s *t*-test (*p* ≤ 0.05) were regarded as the treatment-responsive proteins.

For mass-spectrometry analysis, the differentially expressed protein spots were excised and destained with 50 mM ammonium bicarbonate and dehydrated with 100% acetonitrile (ACN), and then digested with 15 µL of working trypsin (10 ng µL^−1^) (Molecular grade, Promega, Madison, WI, USA) overnight at 37 °C [108]. The final volume was subjected to a mass spectrometer (MALDI MS—ABI Sciex 5800 TOF/TOF System, Massachusetts, USA) controlled by the Flexcontrol 2.4 package using set parameters with the peptide charge of 1+ and peptide tolerance of 150 ppm. Protein identification and quantitative analyses were carried out by submitting MS/MS spectra to the MASCOT server (Matrix Science, London, UK; version 2.2; http://www.matrixscience.com/, accessed on 15 April 2021) [109]. To achieve identification results with high confidence (≥ 95%), protein should have a valuable MOWSE score, sequence coverage greater than 15%, and at least six peptides matched. The available information on identified proteins was assembled with the help of NCBI [110] and Rice Genome Annotation Project (version 7; http://rice.plantbiology.msu.edu, accessed on 27 April 2020) databases. Moreover, the sub-cellular location of identified proteins was assimilated by UniProt databases to understand the protein function [111].

### 4.7. Quantitative Real-Time PCR Analysis

Quantitative real-time (qRT)-PCR analysis was performed to validate ten commonly selected DEPs, using the same lyophilized samples of the six growth stages. Total RNA was extracted from leaves with TRIzol Reagent (Thermo Scientific, Waltham, MA, USA), which was used according to the manufacturer’s instructions. cDNA was synthesized from the total RNA using the BioScriptTM Reverse Transcriptase Reagent kit (Bioline Reagents Limited, London, UK). Amplification of the cDNA fragments of the selected genes was performed on Light Cycler^®^ 480 System (Roche Diagnostics) with iTaq universal SYBR Green supermix (Bio-Rad, USA) and gene-specific primers (Appendix A). An individual reaction contained 10 μL iTaq supermix (Bio-Rad, Hercules, CA, USA), 5 μL of cDNA sample (undiluted), 1 μL of forward and reverse primer mix. Relative expression of genes was calculated by comparing the CT values with the control gene Actin-1 (ACT1) [112] with the 2–ΔΔCT method [113].

### 4.8. Statistical Analyses

Physiological, growth, and biochemical data were collected from three replicates of screening and stage-specific experiments under different N treatments. Data are presented as the mean ± SE (standard error) and were analyzed by one-way ANOVA in the screening experiment (Appendix A) and two-way and three-way ANOVA in the stage-specific experiment (Appendix A) with Minitab Statistical Software (version 2020, LLC, State College, PA, USA). Graphs and heatmaps were drawn in SigmaPlot 12.0 (Systat Software Inc., San Jose, CA, USA) and R-Studio (https://www.rstudio.com/, accessed on 4 January 2021). The significant difference between N treatments of both Panvel and Nagina 22 are indicated separately (* *p* ≤ 0.05; ** *p* ≤ 0.01; *** *p* ≤ 0.001).

## 5. Conclusions

Our comprehensive study of two contrast rice cultivars revealed significant physiological and genetic variations in leaf photosynthesis, the fluorescence of PS-II, plant growth at different growth stages, and grain yields and NUE under low-N supply. Outcomes of the study support that modification in low-efficient plants are required to attain higher physiological growth and crop yields along with improved NUpE, NUtE, and NUE. These alterations are highly regulated at the molecular level and involve key regulators of photosynthetic light reaction and C, N, and protein metabolism at N-limited conditions. Our study further proposes that the fine-tuning of genes involved in these processes provides a promising approach to enhancing NUE at low-N supply. However, NUE is a complex trait in plant species that is probably regulated by multiple genes; single-gene alterations may not be satisfactory for a universal approach to improve NUE under N limitations. Instead, it may be necessary to manipulate multiple gene targets simultaneously. An ammonium transporter and a glutamate synthase were both upregulated in rice using a ‘gene pyramiding method’ to increase NUE [114], which provides little success, but further research is needed. In order to maximize NUE and avoid substrate limitation or end-product inhibition of metabolic pathways, it is necessary to coordinate the modification of leaf N/C metabolic and other associated activities.

## Figures and Tables

**Figure 1 ijms-23-06410-f001:**
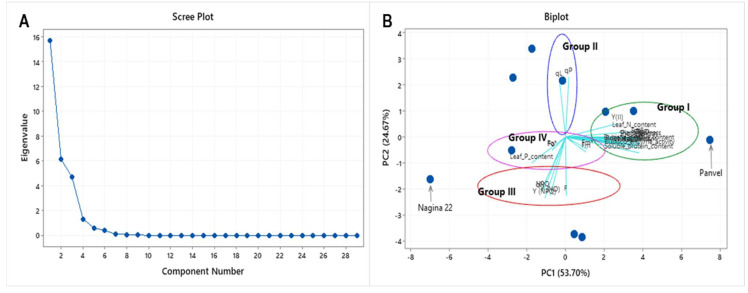
Principle component analysis of 29 response variables of ten rice cultivars under low–NPS supply. (**A**) Scree plot represents the Eigenvalue of 29 traits, and (**B**) biplot represents variance between cultivars and traits. Traits were classified into four groups (group–I, II, III, and IV), and a pair of contrasting cultivars was denoted (Nagina 22 and Panvel).

**Figure 2 ijms-23-06410-f002:**
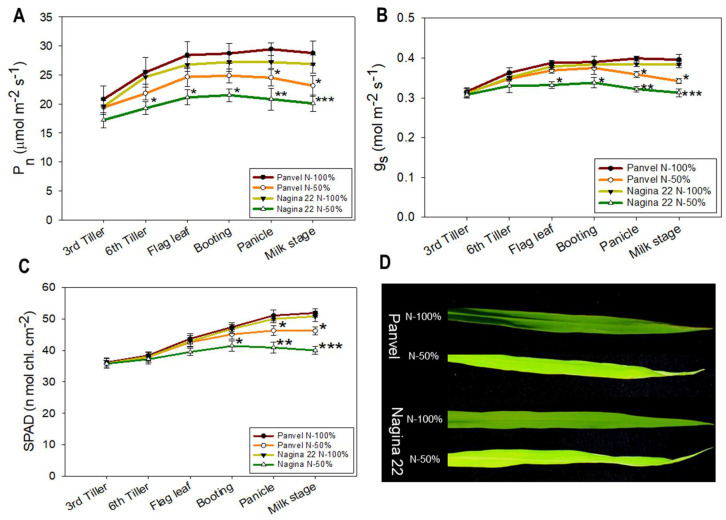
Changes in gas–exchange traits at N–50% treatment in Panvel and Nagina 22 rice cultivars rather than N–100% treatment. P_n_ = net photosynthetic rate (**A**), g_s_ = stomatal conductance (**B**), chlorophyll content measured by SPAD (**C**), and leaves of rice plants at the panicle growth stage (**D**). Each curve point in each graph represents the mean ± SE at each growth stage, and the asterisks denote the significant differences at * *p* < 0.05, ** *p* < 0.01, and *** *p* < 0.001 between the treatments of each cultivar.

**Figure 3 ijms-23-06410-f003:**
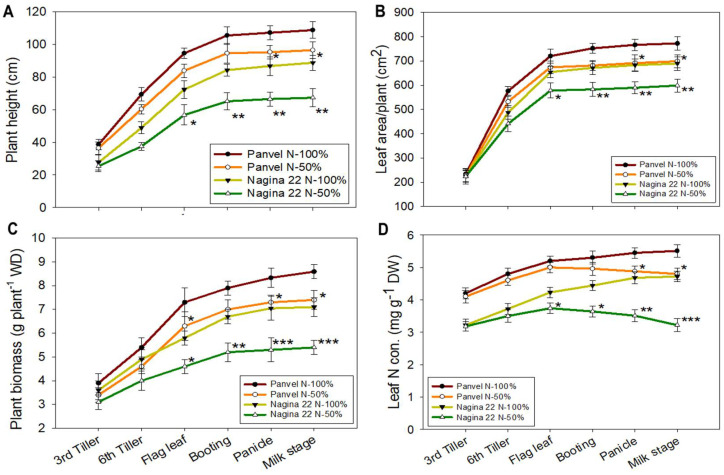
Variation in plant growth and nitrogen uptake of Panvel and Nagina 22 at six growth stages under N–50% and N–100% supply. Plant height (**A**), leaf area per plant (**B**), plant biomass (**C**), and leaf N content were measured in the dry shoot (**D**). Each curve in each graph represents the data set of mean ± SE of each treatment, and the asterisks denote the significant differences at * *p* < 0.05, ** *p* < 0.01, and *** *p* < 0.001 between treatments of each cultivar.

**Figure 4 ijms-23-06410-f004:**
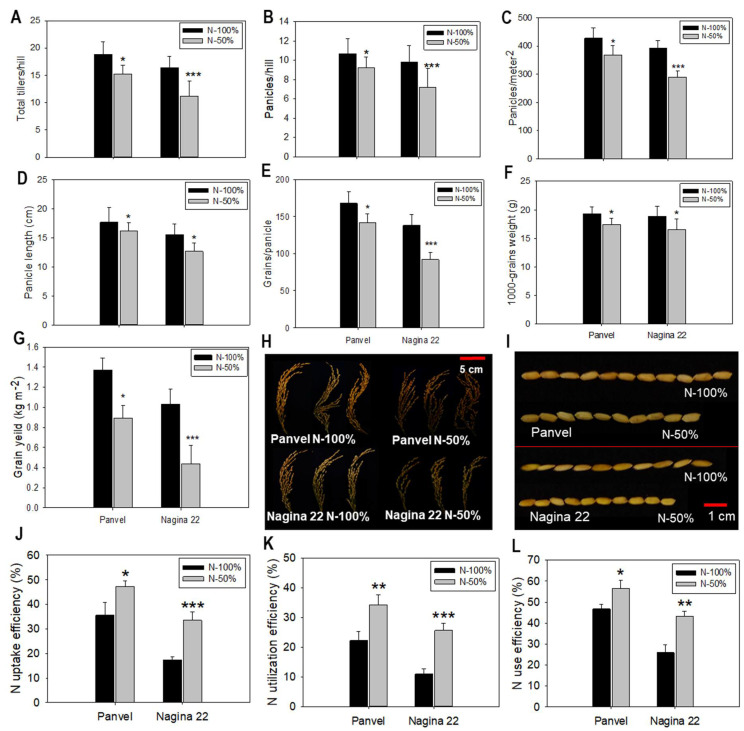
Agronomic and N use efficiency of Panvel and Nagina 22 grown under low N (N–50%) and optimum N (N–100%) supply. Total tillers per hill (**A**), number of panicles per hill (**B**) and number of panicles per meter square (**C**), length of panicles (**D**), number of grains set per panicle (**E**), 1000 grains weight (**F**), grain yield per meter square (**G**), size of panicles (**H**) and ten rice grains (**I**), and percentage of N uptake efficiency (**J**), N utilization efficiency (**K**), and N use efficiency (**L**). Bars in each graph represent the mean ± SE of each treatment, and the asterisks denote the significant differences at * *p* < 0.05, ** *p* < 0.01, and *** *p* < 0.001 between treatments of each cultivar.

**Figure 5 ijms-23-06410-f005:**
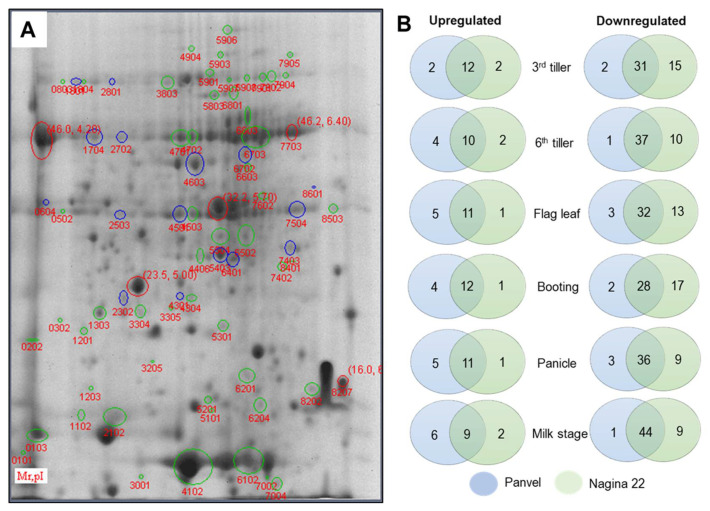
Differentially expressed proteins (DEP) and their variation between Panvel and Nagina 22 at different growth stages under low–N supply. Representation of DEPs on gel map, blue and green circled spots are upregulated and downregulated proteins, respectively; and red circled spots are references for experimental molecular weight and isoelectric point (**A**). Venn diagrams represent the differences of upregulated and downregulated proteins between rice cultivars and growth stages under low-N supply (**B**).

**Figure 6 ijms-23-06410-f006:**
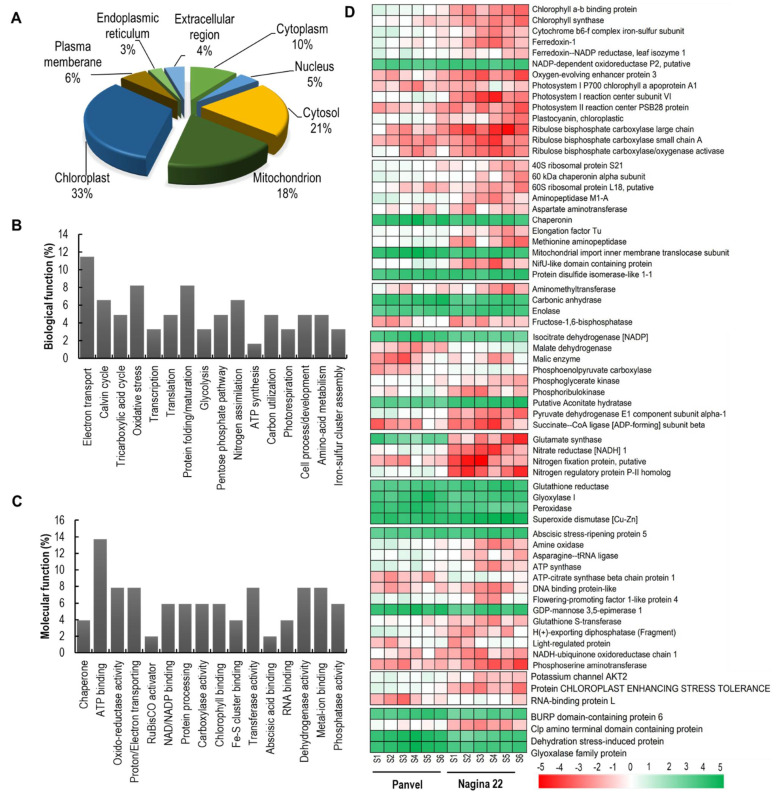
Spatial and functional distributions and heatmap of identified differentially expressed proteins (DEPs) at different growth stages of Panvel and Nagina 22. Cell component contribution of DEPs (**A**) and percentage of DEPs in different biological (**B**) and molecular (**C**) processes. Heatmap of identified DEPs at six specific growth stages (S1–3rd tiller, S–6th tiller, S3–flag leaf, S4–booting, S5–panicle, and S6–milk stages) of rice cultivars Panvel and Nagina 22 (**D**). The fold–change of upregulated protein spot volumes was calculated with treatment/optimum, whereas the fold–change of downregulated protein spot volumes was calculated with optimum/treatment. More than 2–fold relative expressed spots were considered as differentially expressed between treatments.

**Figure 7 ijms-23-06410-f007:**
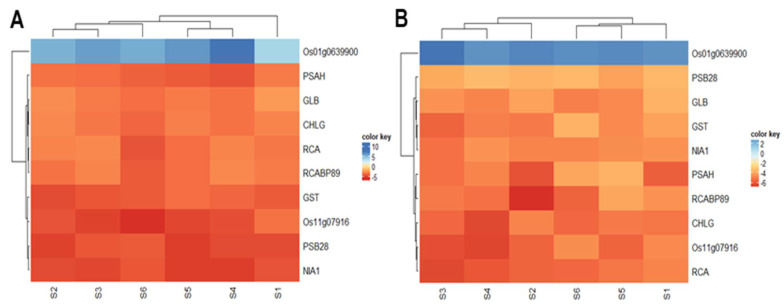
Heatmaps represent the gene expression qRT–PCR by cultivar Panvel (**A**) and Nagina 22 (**B**) under low–N supply. The relative expression was evaluated with the comparative cycle threshold method taking Actin–1 as the reference gene. The relative expression levels were calculated and normalized concerning Actin-1 mRNA (S1–3rd tiller, S2–6th tiller, S3–flag leaf, S4–booting, S5–panicle, and S6–milk stages).

**Figure 8 ijms-23-06410-f008:**
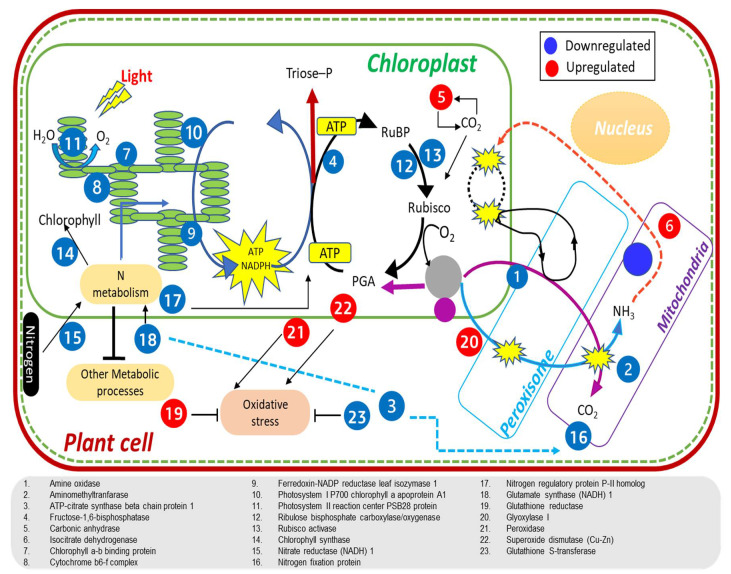
Schematic representation of candidate genes and their differential expression under low–N supply in rice plants. Most of the candidate genes were involved in energy metabolism and photosynthetic processes, but some candidate genes were involved in N metabolism and oxidative defense system. The downregulation of proteins possibly decreased the energy status and assimilatory powers in low-efficient cultivar (Nagina 22) under low–N supply. Besides this, a few proteins were upregulated that probably enhance oxidative defense and energy level in a high-efficient cultivar (Panvel) under low-N supply.

**Table 1 ijms-23-06410-t001:** Chlorophyll fluorescence differences of PSII in rice leaves of Panvel and Nagina 22 at different growth stages under optimum and low-N supply.

Traits	Growth Stages	Panvel		Nagina 22	
		N-100%	N-50%	N-100%	N-50%
**F_v_/F_m_**	3rd tiller	0.796 ± 0.036	0.781 ± 0.032	0.794 ± 0.043	0.735 ± 0.034 *
	6th tiller	0.809 ± 0.041	0.787 ± 0.028	0.801 ± 0.029	0.723 ± 0.042 *
	Flag leaf	0.825 ± 0.037	0.799 ± 0.035	0.807 ± 0.033	0.729 ± 0.038 *
	Booting	0.822 ± 0.044	0.776 ± 0.042 *	0.805 ± 0.061	0.672 ± 0.028 **
	Panicle	0.821 ± 0.041	0.728 ± 0.056 *	0.803 ± 0.054	0.662 ± 0.035 **
	Milk stage	0.821 ± 0.035	0.716 ± 0.049 **	0.806 ± 0.048	0.658 ± 0.046 ***
**Φ_PSII_**	3rd tiller	0.684 ± 0.026	0.671 ± 0.031	0.677 ± 0.026	0.658 ± 0.029
	6th tiller	0.754 ± 0.033	0.722 ± 0.035	0.738 ± 0.025	0.654 ± 0.021 **
	Flag leaf	0.762 ± 0.024	0.682 ± 0.032 *	0.743 ± 0.036	0.632 ± 0.042 ***
	Booting	0.746 ± 0.038	0.668 ± 0.028 *	0.725 ± 0.031	0.605 ± 0.029 ***
	Panicle	0.715 ± 0.041	0.637 ± 0.037 *	0.692 ± 0.035	0.578 ± 0.044 ***
	Milk stage	0.672 ± 0.036	0.605 ± 0.039 *	0.664 ± 0.034	0.527 ± 0.037 ***
**ETR**	3rd tiller	164.55 ± 11.4	161.43 ± 12.1	162.87 ± 10.3	158.30 ± 10.4
	6th tiller	181.39 ± 10.3	173.70 ± 11.7	177.55 ± 12.7	157.34 ± 11.5 *
	Flag leaf	183.32 ± 09.7	164.09 ± 13.4 *	178.75 ± 11.6	152.04 ± 11.9 **
	Booting	179.47 ± 13.5	160.70 ± 12.6 *	174.42 ± 12.9	145.55 ± 12.1 **
	Panicle	172.01 ± 12.8	153.25 ± 11.4 *	166.48 ± 13.2	139.05 ± 10.6 **
	Milk stage	161.67 ± 14.2	145.55 ± 15.3 *	159.74 ± 14.5	126.78 ± 11.2 ***
**qP**	3rd tiller	0.807 ± 0.024	0.798 ± 0.031	0.802 ± 0.025	0.779 ± 0.032
	6th tiller	0.825 ± 0.036	0.812 ± 0.028	0.819 ± 0.029	0.736 ± 0.027 *
	Flag leaf	0.867 ± 0.027	0.801 ± 0.037 *	0.855 ± 0.031	0.758 ± 0.025 **
	Booting	0.832 ± 0.041	0.763 ± 0.043 *	0.821 ± 0.036	0.724 ± 0.032 **
	Panicle	0.801 ± 0.038	0.714 ± 0.041 **	0.793 ± 0.028	0.687 ± 0.044 ***
	Milk stage	0.785 ± 0.023	0.678 ± 0.036 **	0.778 ± 0.042	0.583 ± 0.041 ***
**NPQ**	3rd tiller	0.064 ± 0.011	0.071 ± 0.016	0.068 ± 0.014	0.089 ± 0.015 *
	6th tiller	0.042 ± 0.013	0.053 ± 0.012	0.047 ± 0.011	0.073 ± 0.013 *
	Flag leaf	0.023 ± 0.008	0.033 ± 0.010	0.029 ± 0.010	0.056 ± 0.014 *
	Booting	0.046 ± 0.012	0.054 ± 0.013	0.048 ± 0.013	0.073 ± 0.016 **
	Panicle	0.073 ± 0.019	0.091 ± 0.017 *	0.075 ± 0.020	0.115 ± 0.022 ***
	Milk stage	0.096 ± 0.014	0.125 ± 0.023 **	0.101 ± 0.019	0.168 ± 0.034 ***

Data in the table represent the mean ± SE of each treatment, and the asterisks denote the significant differences at * *p* < 0.05, ** *p* < 0.01, and *** *p* < 0.001 between the treatments of each cultivar.

**Table 2 ijms-23-06410-t002:** Rubisco enzyme activity and Rubisco and soluble protein content in rice leaves of Panvel and Nagina 22 at six growth stages under optimum and low N applications.

Traits	Growth Stages	Panvel		Nagina 22	
		N-100%	N-50%	N-100%	N-50%
Rubisco enzyme activity (n mol^−1^ g^−1^ FW)	3rd tiller	215.9 ± 12.2	203.3 ± 14.3	208.8 ± 11.6	197.5 ± 13.8
	6th tiller	224.2 ± 11.9	207.2 ± 11.5	218.3 ± 13.2	199.9± 12.4 *
	Flag leaf	234.6 ± 13.5	211.2 ± 11.6 *	229.3 ± 12.3	202.3 ± 14.6 *
	Booting	235.4 ± 09.8	209.4 ± 11.3 *	230.8 ± 13.6	198.5 ± 12.8 **
	Panicle	235.7 ± 14.1	204.9 ± 11.5 **	228.2 ± 12.2	186.4 ± 11.6 ***
	Milk stage	234.9 ± 10.2	201.3 ± 13.7 **	225.7 ± 11.2	175.2 ± 12.2 ***
Rubisco protein content (mg m^−2^ FW)	3rd tiller	0.139 ± 0.025	0.134 ± 0.031	0.132 ± 0.027	0.125 ± 0.026
	6th tiller	0.152 ± 0.032	0.141 ± 0.028	0.143 ± 0.021	0.130 ± 0.023 *
	Flag leaf	0.163 ± 0.029	0.148 ± 0.025 *	0.156 ± 0.027	0.138 ± 0.028 *
	Booting	0.167 ± 0.033	0.154 ± 0.027	0.162 ± 0.031	0.141 ± 0.029 **
	Panicle	0.171 ± 0.022	0.155 ± 0.026 *	0.165 ± 0.024	0.140 ± 0.028 **
	Milk stage	0.174 ± 0.027	0.154 ± 0.023 *	0.169 ± 0.025	0.142 ± 0.021 **
Soluble protein content (mg g^−2^ FW)	3rd tiller	29.31 ± 3.23	23.82 ± 2.42	24.56 ± 3.31	20.13 ± 4.11
	6th tiller	32.49 ± 4.21	25.35 ± 3.26	28.78 ± 3.45	24.89 ± 3.27 *
	Flag leaf	35.46 ± 2.78	30.04 ± 3.16	32.16 ± 3.28	27.55 ± 3.14 *
	Booting	36.19 ± 4.03	32.66 ± 4.15 *	33.73 ± 3.79	27.49 ± 3.26 **
	Panicle	35.96 ± 3.99	31.72 ± 3.48 *	33.11 ± 4.12	26.12 ± 3.65 **
	Milk stage	36.11 ± 4.09	30.37 ± 3.67 *	33.51 ± 4.55	24.47 ± 4.13 ***

Data in the table represents mean ± SE of each treatment, and the asterisks denotes the significant differences at * *p* < 0.05, ** *p* < 0.01 and *** *p* < 0.001 between treatments of each cultivar (FW = fresh weight).

## Data Availability

Not applicable.

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
