# Peer review of "Physiological, Agronomical, and Proteomic Studies Reveal Crucial Players in Rice Nitrogen Use Efficiency under Low Nitrogen Supply"

_ijms, 2022, doi:10.3390/ijms23126410_

Round 1
Reviewer 1 Report
The manuscript describes the issue of reduced nitrogen dose to physiological and biochemical rice parameters. This is research that has been monitored for a relatively long time. However, the research is also focused on other parameters that have not yet been monitored in such comprehensive research. However, it is a pity that the correlations between the parameters were not observed. The abstract is sufficient, but it could be supplemented by the doses of nitrogen used and its forms. The literature review is rather more general and is not entirely focused on the issues addressed. The results are described on the basis of graphs and tables. It is a pity that the graphs are small and hard to read. Sometimes the axis descriptions overlap with the values ​​listed on them. I would also add values ​​of statistically significant differences to the twext. The methodology is appropriate. Please complete the location and weather. The summary is adequate. Unfortunately, older literature is cited. I recommend it to be redesigned and supplemented with new literature data.
Author Response
RESPONSES TO THE COMMENTS OF REVIEWER # 1
We thank the esteemed reviewer for critically evaluating the manuscript and for giving valuable suggestions and comments. The responses/clarifications of the comments/suggestions of the reviewers are given below, point-by-point. The changes/corrections have been made by track changes in the manuscript and supplementary files are also updated.
Comment # 1. The abstract is sufficient, but it could be supplemented by the doses of nitrogen used and its forms. The literature review is rather more general and is not entirely focused on the issues addressed.
Response #1: Thanks for your suggestion. The nitrogen doses that were used in the experiment have been added to the abstract. Also, the literature has been improved and focused on issues addressed in recent literature.
Comment # 2. The results are described on the basis of graphs and tables. It is a pity that the graphs are small and hard to read. Sometimes the axis descriptions overlap with the values ​​listed on them. I would also add values ​​of statistically significant differences to the text.
Response # 2: Thanks for your suggestion. The corrections have been made. Most of the graphs have been improved and overlapping issues have been solved. The statistical values have been added to the text where there was a need.
Comment # 3: The methodology is appropriate. Please complete the location and weather.
Response: # 3: Thanks for your suggestion. The location and weather of the experimental site have been completed in the materials and methods section.
Comment # 4. Unfortunately, older literature is cited. I recommend it be redesigned and supplemented with new literature data.
Response # 4: Thanks for your suggestion. The literature has been redesigned and updated with recent citations according to the reviewers’ suggestions.
Reviewer 2 Report
Review Remarks on the manuscript [Physiological, Agronomical and Proteomic Studies Reveal Crucial Players in Rice Nitrogen Use Efficiency Under Low Nitrogen Supply]. Overall, the presentation of the manuscript is good. However, there are some general points and scientific queries below-mentioned needed to be addressed in the manuscript:
· In lines 38 and 39, check the word unutilized.
· In lines 39-40; harming ecosystems; please give some examples.
· In line 86, replace the word higher with an appropriate one.
· In lines 101-102: At the fifth tiller stage, physiological, growth, and biochemical differences were measured in ten rice cultivars under low N, P and S (NPS) conditions; why only the fifth tiller stage was selected? Was there any specific reason was such selection?
· Use this paper: https://dx.doi.org/10.1016%2Fj.tjem.2018.08.001 to represent the interpretation of correlation throughout the manuscript; please use standard terminology like weak, moderate, and strong positive and negative correlation.
· In line 134; a tremendous significant difference; what does tremendous, please use appropriate statistical terminology throughout the manuscript.
· In Fig. 2, please check overlapping text in the A and C parts and modify accordingly.
· In Tables 1 and 2, under N-100 column, results are insignificant, I can understand that results can be insignificant but it would be great if the authors could explain the reasons for such observations.
· In Figure 5, what red circle represents?
· I found that in the discussion part there is mostly the repetition of the results. Undoubtedly, representation is good but couldn’t fulfill its purpose to the fullest. So please club this part with discussion or add more literature to support the findings of the study, whatever is most suitable to the authors.
Author Response
Comment # 1. In lines 38 and 39 checks the word unutilized.
Response #1: Thanks for your suggestion. The word “unutilized” has been replaced with the appropriate word “unused”.
Comment # 2: In lines 39-40; harming ecosystems, please give some examples.
Response # 2: Thanks for your suggestion. Some examples of harming ecosystems have been mentioned in the text. Line # 40-41.
Comment # 3. In line 86, replace the word higher with an appropriate one.
Response # 3: Thanks for your suggestion. The word higher has been replaced with later. Line # 88.
Comment # 4: In lines 101-102: At the fifth tiller stage, physiological, growth, and biochemical differences were measured in ten rice cultivars under low N, P and S (NPS) conditions; why only the fifth tiller stage was selected? Was there any specific reason was such selection?
Response # 4: Thanks for your query. The fifth tiller stage was selected for screening the physiological, growth, and biochemical differences among the ten rice cultivars under low NPS conditions; because at this stage the plants were 5-6 weeks old and responded very well to supplied nutrients for maximizing the tillers. This growth stage enables us to find significant differences between the rice cultivars in most of the traits measured.
Comment # 5: Use this paper: https://dx.doi.org/10.1016%2Fj.tjem.2018.08.001 to represent the interpretation of correlation throughout the manuscript; please use standard terminology like weak, moderate, and strong positive and negative correlation.
Response #5: Thanks. The paper is very useful to understand correlation representation. Standard terminology has been adopted to express the correlation and significance of the data throughout the manuscript.
Comment # 6: In line 134; a tremendous significant difference; what does tremendous, please use appropriate statistical terminology throughout the manuscript.
Response # 6: Thanks for your suggestion. The word tremendous has been removed and a standard statistical terminology has been adopted through out the manuscript.
Comment # 7: In Fig. 2, please check overlapping text in the A and C parts and modify accordingly.
Response # 7: Sorry for the mistake. The figure has been modified according to your suggestion.
Comment # 8: In Tables 1 and 2, under the N-100 column, results are insignificant, I can understand that results can be insignificant but it would be great if the authors could explain the reasons for such observations.
Response # 8: In this study, the effect of nitrogen treatment was more prominent than the changes between the growth stages for the responses mentioned in Tables 1 and 2. So, the significant differences were shown at *P<0.05, **P<0.01, and ***P<0.001 between treatments (N-50 column in comparison to N-100 column) of each cultivar, which is mentioned in the footnote of each table.
Comment # 9: In Figure 5, what red circle represents?
Response # 9: Thanks for the query. The red circle in Figure 5A represents reference spots for experiential molecular weight and isoelectric point (pI) of protein spots in the 2D map. The experimental molecular weight and pI are mentioned in Supplementary Table 8.
Comment # 10: I found that in the discussion part there is mostly the repetition of the results. Undoubtedly, representation is good but couldn’t fulfill its purpose to the fullest. So please club this part with discussion or add more literature to support the findings of the study, whatever is most suitable to the authors.
Response # 10: Thanks for the suggestion. The discussion has been improved and more literature has been added to support the findings of the study.
Reviewer 3 Report
This is and interesting article about study of NUE in contrast rice cultivars under nitrogen-deficient conditions, but there are minor comments.
L.93. Reference to Suppl. Tables and Figures should be moved to the Materials and Methods section.
L.376. Reference to Suppl. Table S9 can be removed as it is mentioned in the Materials and Methods section.
L.517-519. Unfortunately, I did not find glutamate synthase in Table 6d and Suppl. Table S8. Please clarify where this protein is located.
L.520-522. This statement is not entirely correct. Glutamine synthetase, unlike glutamate synthase, is one of the most widely used enzymes for improving NUE via genetic transformation and often demonstrated good results in transgenic plants (Lebedev et al. Genetic engineering and genome editing for improving nitrogen use efficiency in plants. Cells, 2021, 10 :3303).
L.563-566. Why was sulfur given special attention, and not potassium, which is a more important nutrient?
L.737. “Glutamate synthetase” should be corrected to “glutamate synthase”.
Fig. 3c, Suppl. Fig. S2d. Is plant biomass indicated as FW or DW?
Table 2. Rubisco enzyme activity: it is better to leave one sign after the dot.
Please indicate the genetic origin of contrast rice cultivars (Panvel and Nagina 22).
Author Response
We thank the esteemed reviewer for critically evaluating the manuscript and for giving valuable suggestions and comments. The responses/clarifications of the comments/suggestions of the reviewers are given below, point-by-point. The changes/corrections have been made by track changes in the manuscript and supplementary files are also updated.
Comment # 1: L.93. Reference to Suppl. Tables and Figures should be moved to the Materials and Methods section.
Response # 1: Thanks for the suggestion. The reference of the Supplementary Tables S1-S5 and Figures S1, and S2 has been moved to the Materials and methods section. Line # 543-544.
Comment # 2: L.376. Reference to Suppl. Table S9 can be removed as it is mentioned in the Materials and Methods section.
Response # 2: Thanks for the suggestion. The reference to Supplementary Table S9 has been removed in the Materials and Methods section.
Comment # 3: L.517-519. Unfortunately, I did not find glutamate synthase in Table 6d and Suppl. Table S8. Please clarify where this protein is located.
Response #3: Sorry for the mistake. During data transformation glutamate synthase was missed. Now, it has been added to both Fig. 6D and Suppl. Table S8.
Comment # 4: L.520-522. This statement is not entirely correct. Glutamine synthetase, unlike glutamate synthase, is one of the most widely used enzymes for improving NUE via genetic transformation and often demonstrated good results in transgenic plants (Lebedev et al. Genetic engineering and genome editing for improving nitrogen use efficiency in plants. Cells, 2021, 10 :3303).
Response #4: The statement has been checked and modified according to the literature with consideration of the above-mentioned article.
Comment # 5: L.563-566. Why was sulfur given special attention, and not potassium, which is a more important nutrient?
Response # 5: Nitrogen (N), phosphorus (P), and sulfur (S) nutrients have a deep interaction for uptake and metabolism of the nutrients. The optimum N supply establishes uptake of P and provides components like nucleic acids, sugar phosphate, and phospholipids for plant growth and metabolism. On the other hand, S and N have dependent interaction in such a way that the inadequacy of one nutrient reduces the uptake and metabolism of the other nutrient. Whereas the low potassium impacts cation exchange capacity and lowers the concentration and uptake of other nutrient ions in plants. So, the study was focused to find efficient and inefficient rice plants for low N, P, and S, rather than K.
Comment # 6: L.737. “Glutamate synthetase” should be corrected to “glutamate synthase”.
Response #6: Sorry for the mistake. The spelling mistake has been corrected in the text.
Comment # 7: Fig. 3c, Suppl. Fig. S2d. Is plant biomass indicated as FW or DW?
Response #7: The plant biomass was measured in DW (dry weight) and has been indicated in both Fig. 3c and Suppl. Fig. S2d.
Comment # 8: Table 2. Rubisco enzyme activity: it is better to leave one sign after the dot.
Response #8: The modified Table 2 has been added with one sign after the dot in the rubisco enzyme activity.
Comment # 9: Please indicate the genetic origin of contrast rice cultivars (Panvel and Nagina 22).
Response #9: Thanks for your suggestion. The genetic origin of the rice cultivars Pavel and Nagina 22 has been added in the materials and methods section.
Round 2
Reviewer 1 Report
The submitted manuscript is a corrected version of the original manuscript based on reviewers' opinions. The authors did not fundamentally change the original focus of the manuscript, but accepted comments and remarks. The abstract was supplemented with variants of the experiment, which gives the reader a better overview of the issue. I really appreciate the change of the introductory part and the exchange of older literature for current literature. This will improve the possible citation rate of the article. The methodology was supplemented, for example, by the course of the weather, which will improve the informative value of the results. The results were supplemented by a statistical evaluation and at the same time the graphs were adjusted.